# Continuous Change Detection and Classification—Spectral Trajectory Breakpoint Recognition for Forest Monitoring

Yangjian Zhang [1,2,3], Li Wang [1,*], Quan Zhou [1,3], Feng Tang [4], Bo Zhang [1,3], Ni Huang [1] and Biswajit Nath [5]

1   State Key Laboratory of Remote Sensing Science, Aerospace Information Research Institute, Chinese Academy of Sciences, Beijing 100094, China; zhangyangjian19@mails.ucas.ac.cn (Y.Z.); zhouquan201@mails.ucas.ac.cn (Q.Z.); zhangbo20@mails.ucas.ac.cn (B.Z.); huangni@radi.ac.cn (N.H.)
2   School of Electronic, Electrical and Communication Engineering, University of Chinese Academy of Sciences, Beijing 100049, China
3   University of Chinese Academy of Sciences, Beijing 100049, China
4   School of Land Science and Technology, China University of Geosciences, Beijing 100083, China; tangfeng@cugb.edu.cn
5   Department of Geography and Environmental Studies, Faculty of Biological Sciences, University of Chittagong, Chittagong 4331, Bangladesh; nath.gis79@cu.ac.bd
*   Correspondence: wangli@radi.ac.cn

**Abstract:** Forest is one of the most important surface coverage types. Monitoring its dynamics is of great significance in global ecological environment monitoring and global carbon circulation research. Forest monitoring based on Landsat time-series stacks is a research hotspot, and continuous change detection is a novel approach to real-time change detection. Here, we present an approach, continuous change detection and classification-spectral trajectory breakpoint recognition, running on Google Earth Engine (GEE) for monitoring forest disturbance and forest long-term trends. We used this approach to monitor forest disturbance and the change in forest cover rate from 1987 to 2020 in Nanning City, China. The high-resolution Google Earth images are collected for the validation of forest disturbance. The classification accuracy of forest, non-forest, and water maps by using the optima classification features was 95.16%. For disturbance detection, the accuracy of our map was 86.4%, significantly higher than 60% of the global forest change product. Our approach can successfully generate high-accuracy classification maps at any time and detect the forest disturbance time on a monthly scale, accurately capturing the thinning cycle of plantations, which earlier studies failed to estimate. All the research work is integrated into GEE to promote the use of the approach on a global scale.

**Keywords:** continuous change detection and classification; Landsat time series; forest dynamics monitoring; spectral trajectory breakpoint recognition

## 1. Introduction

Forests are the most extensive vegetation type with the most extensive coverage area, broadest distribution, most complex compositional structure, richest biodiversity, and highest primary productivity [1–3]. Forest can improve the quality of the human living environment and provide habitat for organisms. More importantly, it plays a vital role in the global carbon cycle [4]. Monitoring of forest dynamics is significant for regional ecosystem and climate change research, forest resource investigations, forestry development, and forestry policy-making. It is mainly reflected in the following two aspects. Firstly, forest variation information is essential for terrestrial ecological carbon storage estimation, regional climate change research, and regional ecosystem stability assessment [5,6]. Secondly, real-time monitoring of forest distributions and their dynamic changes is informative to regional ecological environment restoration and reconstruction projects and scientific evaluation of the performance of forest protection projects [7]. Landsat image archives, as

the image data set with the longest duration of Earth observation, play an important role in forest monitoring [8,9].

In general, the existing forest monitoring algorithms based on Landsat time series (LTS) can be categorized into three types [10]: (1) multidate comparison by classification or difference; (2) spectral trajectory-based analysis; (3) the continuous change detection method.

Early multidate comparison methods included image differencing, principal component analysis, tasseled cap transform, post-classification comparison, and change vector analysis [11–13]. The ideas behind these methods are that changes are detected by comparing the difference between images taken at two different time phases. Nevertheless, it is difficult to set suitable thresholds for acquiring high-accuracy change detection results using these methods, and the empirical threshold can not be transferred to other study areas characterized by different vegetation types and vegetation densities. Another disadvantage is that these methods cannot meet the needs of analyses of forest dynamics and change processes [14]. To minimize the spectral differences caused by intra-annual variation in phenology and sun angle differences, the multidate compositing and classification methods are applied [15–18]. One of the advantages is that these methods can include a broad suite of features such as Landsat bands, their derivatives, as well as auxiliary information such as nighttime data, precipitation and tree cover, etc., for classification.

However, these methods can only capture the forest deforestation and forest degradation information on a yearly scale, and the selection of classification features to achieve high accuracy is also a problem. In addition, these methods are not suitable for the direct continuous change process analysis.

The spectral trajectory-based analysis methods have been proposed to meet the needs of changing process analysis. Such algorithms included the vegetation spectral change tracking algorithm (VCT) [19], vegetation regeneration and disturbance estimates through time (Verdet) [20], Landsat-based detection of trends in disturbance and recovery (LandTrendr) [21] algorithms, etc. The VCT algorithm is a threshold-based spectral trajectory algorithm that uses a parameter called the integrated forest z-score (IFZ) for time series analysis. However, the empirical threshold was set according to the regional forest types and forest densities, and the VCT algorithm can not be directly extended to global forest monitoring. The Verdet and LandTrendr are trajectory segmentation algorithms by temporally dividing the time series into differently sloped segments, and each segment corresponds to a different vegetation change state. These algorithms have been proven to can be applied in different geographical regions.

The common feature of the multidate compositing and classification and the spectral trajectory-based analysis methods is that they minimize the seasonal variation and solar altitude differences by reducing the time series to a single image for each year, typically from a date near the peak of the growing season, using either best pixel approaches or another [22–24]. However, these two kinds of algorithms also have two obvious deficiencies. One is that mutation timing is hard to be captured accurately, and the detected moment of change is usually delayed by more than one year. Another is that the information on surface changes within the year is always ignored [14,25]. Hence, such change monitoring algorithms are not continuous change detection algorithms in the true sense.

The continuous change detection and classification (CCDC) algorithm is representative of the continuous change detection algorithms. It was proposed by Zhu Zhe using a harmonic model to fit a time series of spectral information, which minimizes the differences due to seasonal variation and makes use of the various characteristics of the spectrum in the time interval [26]. Combined with statistical breakpoint detection methods and random forest classifiers, the CCDC algorithm can achieve the goal of acquiring land cover classification results at any time, so it also has the advantages of classification comparison methods. However, the original CCDC algorithm has two disadvantages. One of the disadvantages is the selection of the multiply classification features for high accuracy. Another is that the precise abrupt change time at a month scale can not be acquired.

In general, the classification comparison methods, the spectral trajectory analysis methods, and the continuous change detection methods all have their own advantages. The classification methods can easily acquire the yearly classification maps, which is beneficial to forest cover rate analysis year by year. The spectral trajectory analysis methods can be helpful to change process analysis [27]. The continuous change detection methods can be more effective to detect the change during the year, and no need for empirical thresholds. Thus, this paper developed a new algorithm for forest monitoring by combining the advantages of three types of change detection methods.

In this study, disturbance indicated all kinds of forest loss (land cover conversion from a forest land cover to non-forest land cover). The objectives of our study are: (1) to monitor forest disturbance on a monthly scale and improve the classification accuracy of the CCDC algorithm; (2) to map forest classification and estimate forest cover rate each year. Here, we developed a new approach to continuous change detection and classification—spectral trajectory breakpoint recognition (CCDC-STBR). The new algorithm integrates the CCDC algorithm and spectral trajectory breakpoint recognition method. The CCDC algorithm is mainly used to monitor the change of land cover area year by year, while STBR is mainly used to monitor the disturbance of long time series at a monthly scale that current studies failed to estimate. For landcover classification, the optima subset selection method was used to achieve higher classification accuracy than the original CCDC algorithm. For spectral breakpoint recognition, we applied the continuous time series fitted by the harmonic mode, multiply indexes are adopted to recognize the breakpoints in the time-series trajectory to improve the confidence of forest disturbance monitoring, and the change magnitude factor is defined to estimate the disturbance degree. The morphological closing operation and the unsupervised clustering method are also applied to eliminate salt and pepper noises. CCDC-STBR runs on Google Earth Engine, which can generate yearly classification maps and monitor forest disturbance on a monthly scale. It is easy to be adjusted and applied to other regions.

We take Nanning as the case study of forest monitoring to extend this approach to a global scale, and forest disturbance and classification maps are produced for each year from 1987 to 2020. All the monitoring work in this paper was integrated into the GEE platform, and all the related code would be opened to promote the use of this approach on a global scale.

## 2. Materials and Methods

### 2.1. Study Area

Located to the south of Guangxi ($22°12'$–$24°2'$ N, $107°19'$–$107°37'$ E), Nanning is at the junction of south China, southwest China, and the Southeast Asia economic circle (Figure 1). It is also at the intersection of the Pan-Beibu Gulf Economic Cooperation Circle, Greater Mekong Subregion Cooperation Circle, and Pan-Pearl River Delta Cooperation Circle. In recent years, the government of Guangxi Province has also attached importance to ecological environment construction as part of its local economic development. The implementation of ecological restoration, a key type of ecological project, has also gained certain achievements. Nanning has a warm climate, abundant rainfall, and distinct dry and wet conditions, which are especially suitable for forest growth. It is also known as a "Green City", having the most extensive artificial forest plantation area of any prefecture-level city in Guangxi Province. It is also the key demonstration area for afforestation, artificial pure forest transformation, and rocky desertification integrated rehabilitation by the Guangxi Autonomous Region Government (Figure 1). The research area measures 22,100 km$^2$ and is covered by eight Landsat tiles.

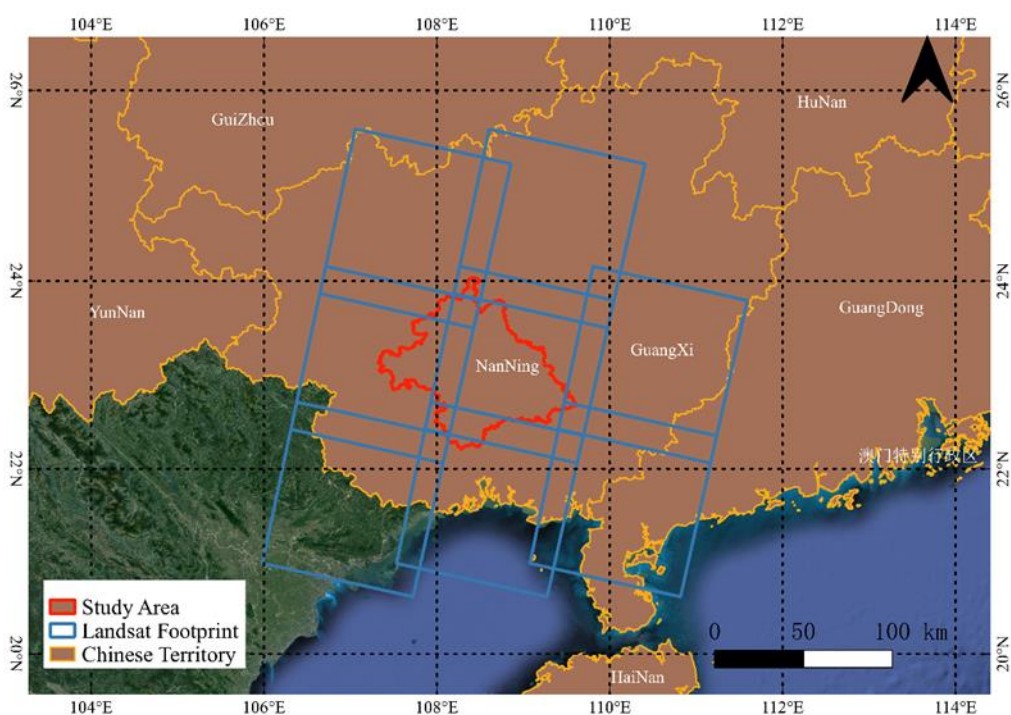

**Figure 1.** Location of the study area.

### 2.2. Data Source and Preprocessing

The Landsat series of satellites have observed the Earth for nearly 50 years. From 1980 to 2020, the data set has gone through four generations of satellites: Landsat 4, 5, 7, and 8 [28]. The sensors corresponding to these four satellites have similar spectral bands and wavelengths, and spatial resolution. At present, the Landsat time series data set has been integrated into the GEE platform, and only a small amount of code can be used to complete the preprocessing [29]. There are two existing Landsat surface reflectance data sets: (1) Landsat Collection 1 Surface Reflectance and (2) Landsat Collection 2 Surface Reflectance. This paper used data from Collection 1. Cloud, cloud shadows, water, and snow were masked by using the pixel_qa, radsat_qa, and sr_aerosol quality bands. From 1987 to 2020, 4053 Landsat images are available for Nanning City. The number of Landsat images available each year is generally more than 30. The average annual cloud cover rate is <50%, which meets the demands of continuous change detection. (Figure 2, Table 1). In addition, this paper also collected high-definition images from Google Earth and forest masks of Nanning City from 2000 for land classification and forest change detection.

**Table 1.** Original data sets used in this study.

| Data | Description | Source |
|---|---|---|
| Landsat 4, 5, 7, 8 | 4053 Landsat surface reflectance images of Nanning City, with all water and snow pixels masked. Spatial resolution = 30 m. Seven bands were used for change detection: NIR, Red, Green, Blue, Swir1, Swir2, and Temp | Google Earth Engine Data Catalog |
| High-resolution images | For verification of land surface change detection results and Landsat image classification | Google Earth |
| Global Forest Change | Nanning Forest Mask in 2000 | Google Earth Engine Data Catalog |

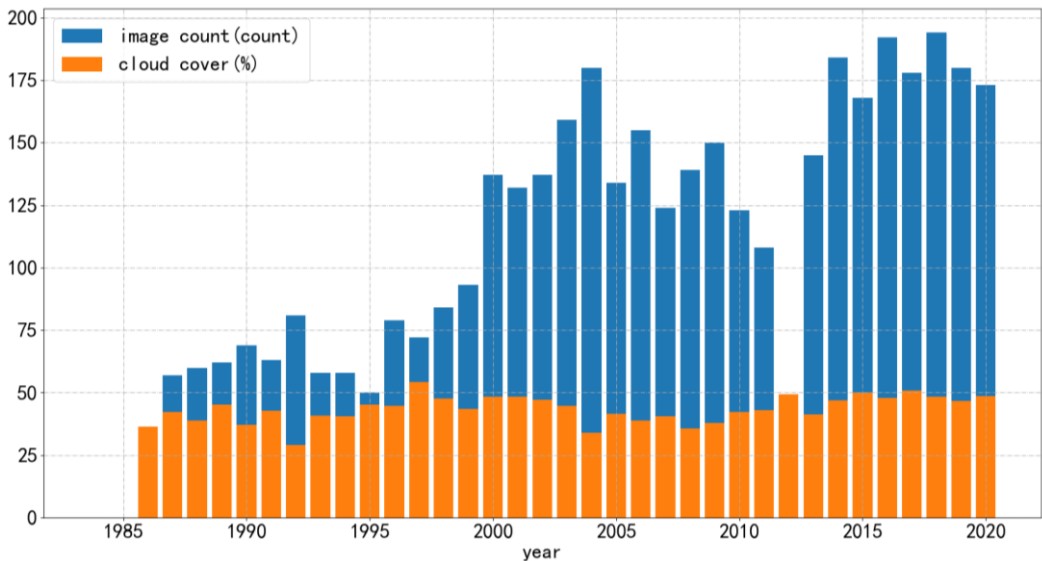

**Figure 2.** Annual count and mean cloud cover percentage of acquired Landsat images.

*2.3. Research Framework*

The application of the CCDC-STBR algorithm to forest monitoring can be divided into four steps (Figure 3): (1) selection of breakpoint recognition bands or detection indexes; (2) time series fitting with a harmonic model; (3) land cover classification and mapping; and (4) forest disturbance detection by breakpoint detection method.

Among them, the first three steps are mainly used to monitor the change of land cover area, and the last step is used to detect the disturbance. In the first three steps, we adopt the optimal subset selection method to choose the best feature set for higher classification accuracy than the original CCDC algorithm.

For the last step, we applied a new breakpoint recognition method to detect the disturbance based on the harmonic fitted time series. The following improvements are made:

1. According to the characteristics that forest disturbance is usually accompanied by an increase in soil composition and a decrease in greenness, the normalized difference fraction index (*NDFI*) and soil index based on the spectral mixture analysis (SMA) model was adopted. The normalized burn ratio (NBR) and the normalized difference vegetation index (NDVI) index are also used for breakpoint identification, for they are frequently used in forest disturbance detection [30,31]. That is, only when the multiply features have changed the pattern is judged as a forest disturbance event;

2. To further improve the disturbance detection accuracy and precisely record the change time at a monthly scale, the continuous harmonic fitting segments are used to extract the disturbance time. The sum confidence was defined as the sum value of the confidences of NDVI, NBR, and *NDFI* (Equation (4)). For every breakpoint in the yearly time-series curve fitted by the harmonic model, only the breakpoint with the highest sum confidences of NBR, NDVI, and *NDFI* was extracted. To eliminate salt and pepper noises, the morphological closing method and the superpixel clustering algorithm are adopted.

Finally, the sum confidence, yearly forest-loss detection results, and yearly classification results were separately acquired (Figure 3). The sum confidence map was used to evaluate the disturbance degree, the loss year detection result was used to detect the disturbance characteristics, and the yearly classification maps were used to monitor forest development in Nanning.

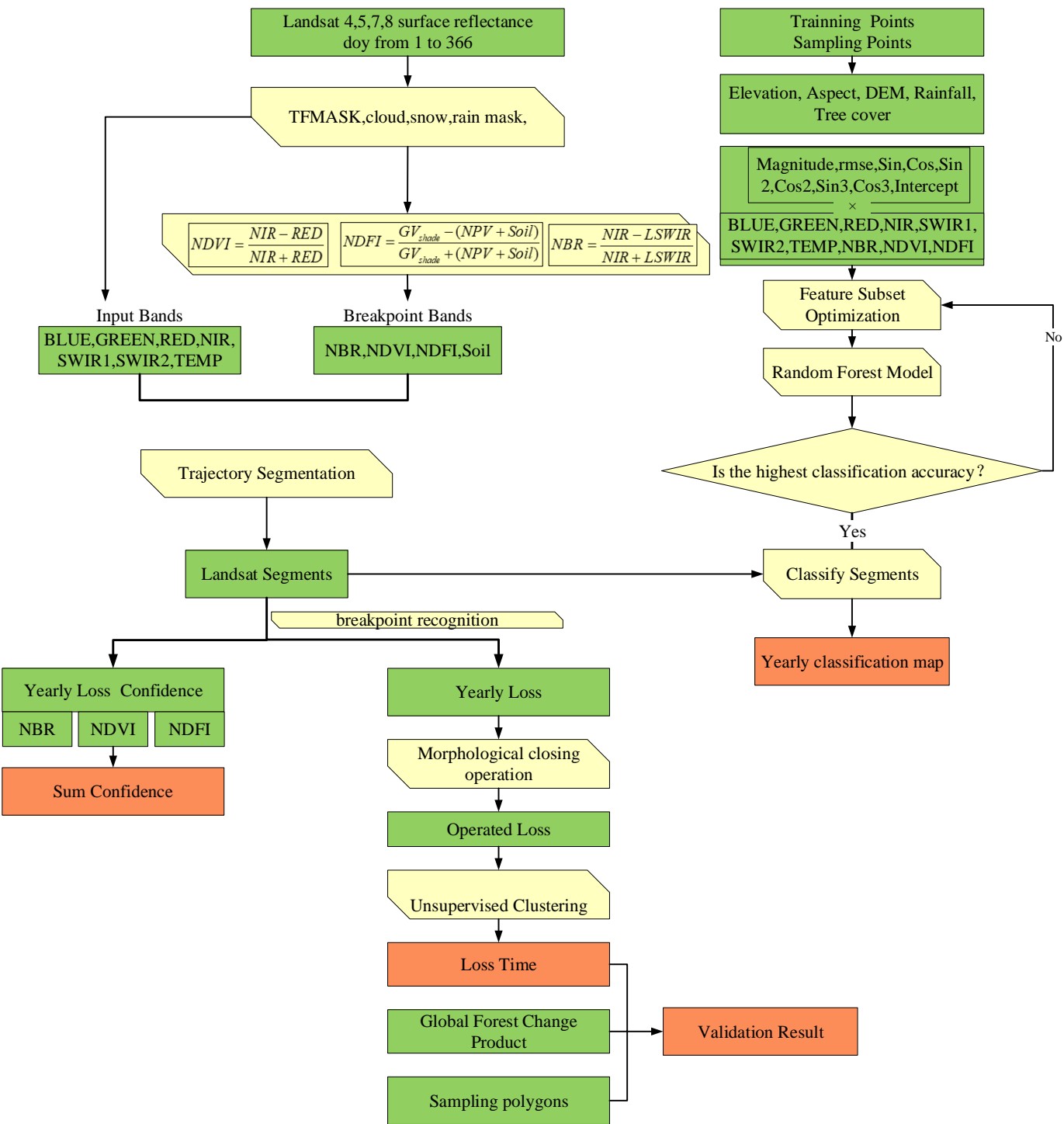

**Figure 3.** Overview of processing steps. The sum confidence result, loss year detection result, and yearly classification maps were separately generated from the Landsat segments. The sum confidence is used to evaluate the magnitude of forest disturbance, the loss year map is helpful for disturbance year evaluation, and the yearly classification map is useful for evaluating the forest cover rate and spatial forest distribution.

### 2.3.1. Forest Monitoring Indexes

The NBR and NDVI indexes are commonly used in forest monitoring. Here, the *NDFI* index was chosen for its higher sensitivity to forest disturbance, and correlated research has proved the *NDFI* is more sensitive to slight forest disturbance such as forest degradation.

Souza et al. proposed the *NDFI* and successfully applied it to the monitoring of tropical forest degradation [32]. The Landsat pixels can be decomposed into fractions of green vegetation (*GV*), non-photosynthetic vegetation (*NPV*), soil, and *shade* through SMA. The SMA model assumes that the pixels can be represented by linear functions of four types of end members: *shade*, soil, greenness (*GV*), and non-photosynthetic vegetation abundance (*NPV*) [33]. Greenness and *shade* should be normalized before constructing the *NDFI* index (Equation (1)), and the *NDFI* index is composed of the normalized value of *GV* and the summed value of *NPV* and soil (Equation (2)).

$$GV_{shade} = \frac{GV}{100 - Shade} \tag{1}$$

$$NDFI = \frac{GV_{shade} - (NPV + Soil)}{GV_{shade} + (NPV + Soil)} \tag{2}$$

The value of *NDFI* ranges between $-1$ and 1. In theory, pure forest pixels have higher greenness and canopy shadow values. The value of *GV* is higher, and the summed value of *NPV* and soil is lower, so the higher the forest coverage, the higher the *NDFI* value. After forest disturbance, the *NPV* and soil values increase significantly, and the *NDFI* value decreases significantly, so the *NDFI* index is sensitive to forest disturbance detection.

### 2.3.2. Breakpoint Detection and Time Series Segmentation Fitting

The CCDC algorithm uses a harmonic model with variable coefficients to fit and predict each spectrum or index of Landsat data on a specified date. The harmonic model has three patterns (four, six, and eight parameters), and the corresponding minimum numbers of observations are 12, 18, and 24 (Equation (3)). When a model-fitted predicted value differs greatly from an actual (observed) value, an abnormal slope occurs or if the first or last observed values deviate from the model-predicted value by three standard deviations during model initialization, the point will be identified as a breakpoint.

$$
\begin{aligned}
\rho(i,t)_{simple} &= a_{0,i} + b_{0,i} \times x + a_{1,i} \times \cos\left(\tfrac{2\pi}{T}x\right) + b_{1,i} \times \sin\left(\tfrac{2\pi}{T}x\right) \\
\rho(i,t)_{simple} &= a_{0,i} + b_{0,i} \times x + a_{1,i} \times \cos\left(\tfrac{2\pi}{T}x\right) + b_{1,i} \times \sin\left(\tfrac{2\pi}{T}x\right) + a_{2,i} \times \cos\left(\tfrac{4\pi}{T}x\right) + b_{2,i} \times \sin\left(\tfrac{4\pi}{T}x\right) \\
\rho(i,t)_{simple} &= a_{0,i} + b_{0,i} \times x + a_{1,i} \times \cos\left(\tfrac{2\pi}{T}x\right) + b_{1,i} \times \sin\left(\tfrac{2\pi}{T}x\right) + a_{2,i} \times \cos\left(\tfrac{4\pi}{T}x\right) + b_{2,i} \times \sin\left(\tfrac{4\pi}{T}x\right) \\
&\quad + a_{3,i} \times \cos\left(\tfrac{6\pi}{T}x\right) + b_{3,i} \times \sin\left(\tfrac{6\pi}{T}x\right)
\end{aligned} \tag{3}
$$

The CCDC algorithm divides the time series of images into a limited number of segments according to the breakpoints in the time series. Each segment has three kinds of coefficients: the harmonic model's fitting coefficients, the spectral phase coefficients, and the interval indicator coefficients (Table 2). The spectral phase coefficients that characterize their seasonal changes are different for different landcover types.

**Table 2.** Segment coefficients.

| Coefficients | Options | Description |
|---|---|---|
| Harmonic coefficients | Sin, Cos, Sin2, Cos2, Sin3, Cos3, Slope, Intercept | Parameters of the harmonic model |
| Derivatives coefficients | AMPLITUDE, PHASE, AMPLITUDE2, PHASE2, AMPLITUDE3, PHASE3, *RMSE*, Magnitude | Seasonal metrics extracted from the harmonic model |
| Interval coefficients | tStart, tBreak, tEnd | Segment indicators |

### 2.3.3. Landcover Classification by Feature Subset Optimization

The features that can be applied to landcover classification in this study can be divided into three categories (Table 3). (1) Band features, which mainly include seven bands: blue, green, red, Nir, Swir1, Swir2, and temperature; (2) index features, which are mainly two normalized indexes (NDVI, NBR) and three indexes (greenness, brightness, wetness) derived from tasseled hat transformation, *NDFI*, and four endmembers, including *shade*, soil, *GV* and *NPV*; (3) auxiliary data features (elevation, aspect, DEM, rainfall, tree cover).

Except for the auxiliary data features, the other features all have the time series parameter features shown in Table 2 after being segmented by the harmonic model. To improve the classification accuracy as much as possible, the wrapper feature selection method is applied to the above features to select the optimal feature subsets. The feature subsets with the highest classification accuracy are screened out.

**Table 3.** All classification features used for subset selection.

| Band | Index Features | Auxillary Features |
|---|---|---|
| Blue, Green, Red, Nir, Swir1, Swir2, Temperature | *NDFI*, NDVI, NBR, Greenness, Brightness, Wetness, *GV*, *NPV*, *Shade*, Soil | Elevation, Aspect, DEM, Rainfall, Tree cover |

The CCDC algorithm requires stable ground sampling points as a training data set for land cover classification. Using the 30 m forest mask, 30 m GlobeLand 30 products, and high-definition images of Nanning City from Google Earth, sampling points were obtained: a total of 540 for forest land, 147 for water bodies, and 402 for other land types (including bare land, cultivated land, grassland, shrubland, wetland, artificial surface, etc.). Land type sampling points were all from areas with stable land cover.

In this study, wrapped feature selection indicates that when the blue, green, red, Nir, Swir1, Swir2, temperature bands, *NDFI*, NDVI, and NBR are selected, the classification accuracy of the random forest classifier is the highest. When *GV*, *NPV*, *shade*, soil, greenness, brightness, wetness, etc., are added, the accuracy of land classification is no longer improved but is, in fact, decreased. Among the auxiliary data features, the rainfall (Rainfall) feature contributes the most to classification accuracy, while terrain factors (elevation, aspect, DEM (digital elevation model)) and tree cover make extremely limited improvements to classification accuracy. Finally, seven band features (blue, green, red, Nir, Swir1, Swir2, temperature), two normalized index features (NDVI and NBR), terrain factor features (elevation, aspect, DEM), rainfall (rainfall), and tree cover are selected as the inputs of the random forest classifier. The corresponding timing parameters of the bands and indexes features are "intercept", "slope", "rmse", "phase", "amplitude", "phase2", "amplitude2", "cos", "sin", "cos2", and "sin2".

2.3.4. Forest Disturbance Detection Based on Spectral Trajectory Breakpoint Recognition

There are two forest disturbance types: (1) forest degradation caused by natural or human factors (such as fire, artificial selective logging, etc.) and (2) severe disturbance, which usually refers to deforestation, forest burned by fire, and transformation into bare land or construction areas.

Forest disturbance events can be detected based on landcover classification results, which cannot obtain an accurate mutation time (precisely capture the change time at a month scale), such as the global forest change (GFC) product. Therefore, this paper adopted the breakpoint recognition method based on the harmonic fitting segments to detect the accurate change time. For breakpoint identification, when the average difference between the first or last observation and the predicted value of the k bands on the segmented time series is greater than three standard deviations or when the average slope of the k bands is >1 (Equation (5)), the point will be identified as a breakpoint. A breakpoint changing from high to low corresponds to a forest-loss event. To acquire the breakpoint with the highest confidence, confidence was defined to measure the magnitude of change of the breakpoint. The magnitude of the breakpoint can be defined as the mean of the five observations made after the mutation event minus the mean of the five observations made before the mutation when the forest-loss event occurred. Magnitude can be recorded as *Mag*, and the confidence can be represented as Equation (4).

$$Confidence = \frac{Mag - RMSE}{Mag + RMSE} \tag{4}$$

$$\frac{1}{k}\sum_{i=1}^{k}\frac{|\rho(i,x_1)-\rho(i,x_1)_{OLS}|}{3\times RMSE_i} > 1 \ or \ \frac{1}{k}\sum_{i=1}^{k}\frac{|c_{1,i}(x)|}{3\times RMSE_i} > 1 \ or \ \frac{1}{k}\sum_{i=1}^{k}\frac{|c_{1,i}(x)|}{3\times RMSE_i/t_{model}} > 1 \quad (5)$$

This paper calculates the sum of the confidence values of the three indices, *NDFI*, NDVI, and NBR, to evaluate the forest disturbance intensity. In addition, for every year, we define the breakpoint with the highest sum confidence value as the disturbance to improve the detection accuracy. Because the identified disturbance still has some noises, combined with the morphological pixel processing method, the morphological closing operation is performed on the breakpoint detection result, and some patches are repaired. Finally, the unsupervised clustering method is used to cluster the morphologically processed images to eliminate salt and pepper noise, and forest-loss disturbance maps are produced for each year from 1987 to 2020.

### 2.3.5. Verification and Comparison of Forest Disturbance Detection

Sampling polygons from high-definition Google Earth images were collected to verify the accuracy of the forest disturbance map. A total of 20,875 sampling points of forest loss have been collected from Google Earth. The accuracy of the disturbance results is evaluated based on the number of *changed pixels* detected on the disturbance map and the total number of *pixels* in the polygon. The *overall accuracy* can be defined as the ratio of the number of *changed pixels* detected to the total number of *pixels* in the polygon (Equation (6)).

$$overall \quad accuracy = \frac{changed \quad pixels}{total \quad pixels} \quad (6)$$

The existing global forest change product can provide the latest time node of forest-loss events. A forest disturbance map of GFC products from 2016 to 2020 was extracted and compared with the forest disturbance map for the same period. The sampling points are used to evaluate the accuracy.

## 3. Results

### 3.1. Landcover Classification Results

For sampling points used for landcover classification, the divisions of the test set and training set are dependent on a random factor. A total of 100 groups of test and training sets were obtained after dividing them by 100 different random factors. The *overall accuracy* and kappa coefficient of the classification were evaluated for each of the 100 groups and averaged separately to evaluate the classification results. The average overall land classification accuracy reached 95.99%, and the average kappa coefficient reached 0.93 (Table 4). Using a random factor of 29, the overall classification accuracy reached 95.16%. A confusion matrix of the landcover classification is shown in Table 4. In this case, there was no misclassification of forest landcover and no misclassification of other landcover types as forest land. Classification confusion mainly occurred between water and other land types. Finally, this study used the trained random forest model to classify the Landsat segment stage-by-stage. According to the stage-by-stage classification results, a landcover classification map of any year could be obtained.

**Table 4.** Confusion matrix of CCDC land cover classification.

| *Overall Accuracy = (167 + 39 + 109)/(167 + 39+9 + 7 + 109) = 95.16% Kappa Coefficient = 0.93* | | | | | |
|---|---|---|---|---|---|
| Class | Forest | Water | Other | Total | User's Accuracy |
| Forest | 167 | 0 | 0 | 167 | 100% |
| Water | 0 | 39 | 7 | 46 | 81.25% |
| Other | 0 | 9 | 109 | 118 | 93.97% |
| Total | 167 | 48 | 116 | 331 | |
| Producer accuracy | 100% | 84.78% | 92.37% | | |

Figure 4 shows forest and water distribution maps of Nanning in eight periods: 1987, 1990, 1995, 2000, 2005, 2010, 2015, and 2020. The spatial distribution maps show that forests were mainly distributed in the north, northwest, and central parts of Nanning but rarely in the south, while there were more water bodies in the south, which is closely related to the abundant rainfall of Nanning. In general, the overall forest distribution pattern in Nanning is very stable. Over nearly three decades, urban change, urban construction, and urban planning in Nanning have not affected the forest distribution pattern, and the forest coverage and area have been increasing (Figures 5 and 6), from 45.5% in 1987 to about 49% in 2020. From 1987 to 2005, Nanning's forest coverage showed slow linear growth with an average annual growth rate of 0.074%. After 2005, Nanning's forest coverage showed more significant growth, with a linear growth rate of about 0.214%. Over the whole period from 1987 to 2020, Nanning's forest coverage increased significantly, with an average annual growth rate of 0.123%.

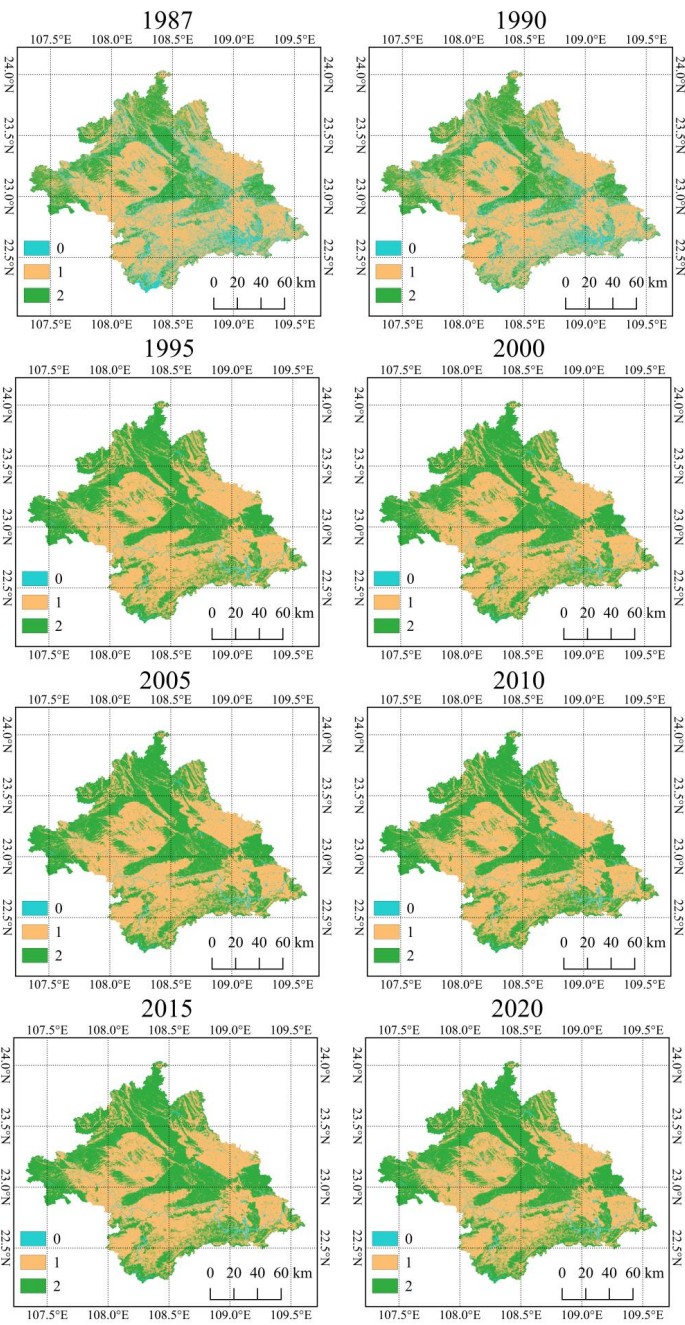

**Figure 4.** Landcover classification maps of Nanning in eight years between 1987 and 2020.

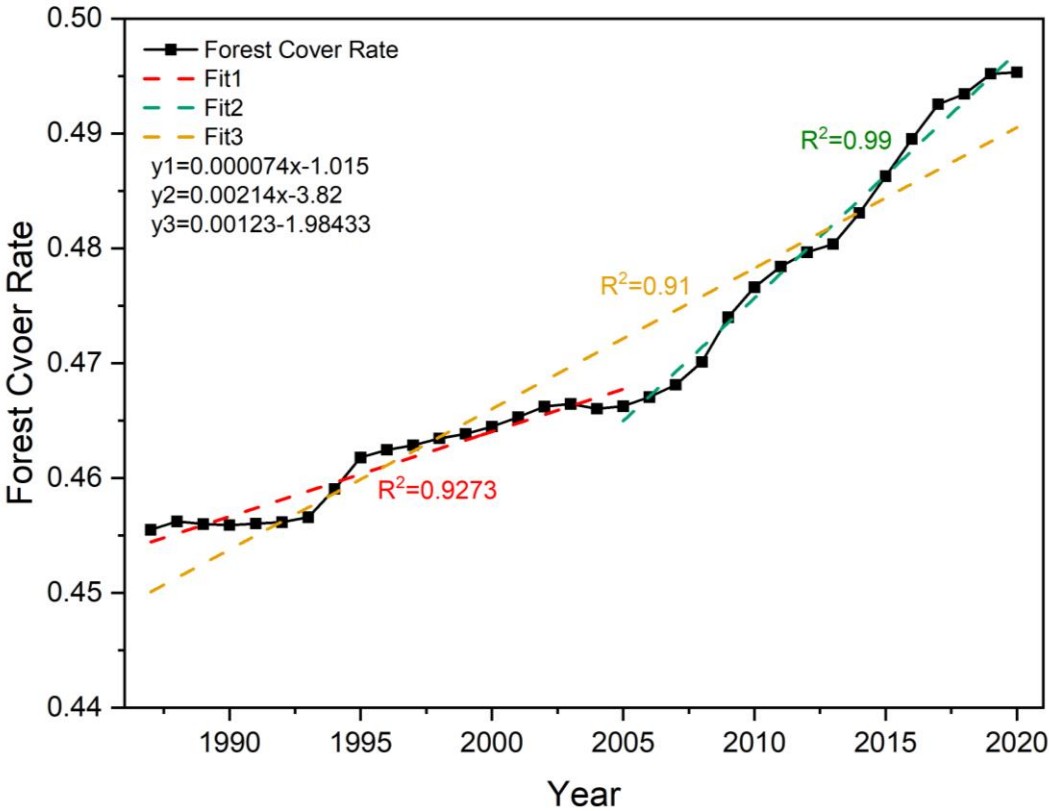

**Figure 5.** Forest coverage changes in Nanning from 1987 to 2020.

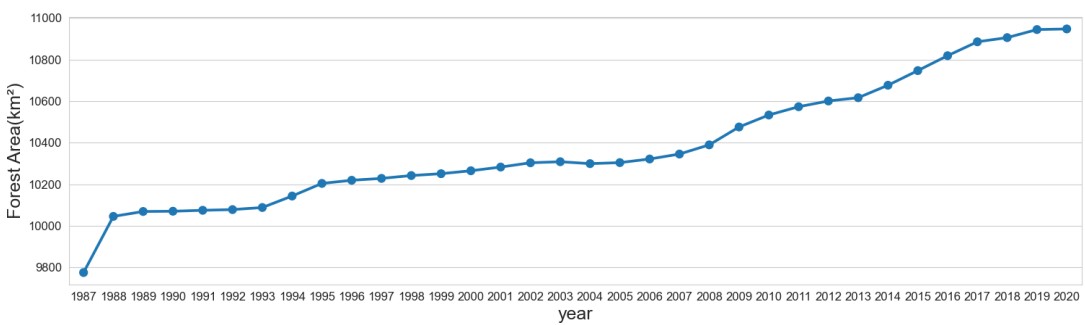

**Figure 6.** Changes in forest cover area in Nanning from 1987 to 2020.

The changes in forest coverage are closely related to the Nanning government's policies of improving forest quality and planning and regulating forest tree species and planting patterns. Since 2008, the Nanning government has implemented forest protection and transformation projects with the goal of creating a national forest city and building a forest ecosystem around the city [34]. These policies have had significant impacts on the growth of forest areas in Nanning.

Based on annual forest distribution maps of Nanning City from 1987 to 2020, the annual forest coverage areas were calculated. From 1987 to 2020, the forest coverage area continued to rise and peaked in 2020 at 10,946 km$^2$. In 1987, the forest coverage area was the lowest. From 1988 to 2008, the forest coverage area increased steadily and slowly, with an average annual growth rate of 0.169% and an average annual growth area of 1600 ha. After 2008, forest coverage increased rapidly. The yearly growth rate was significantly higher than before, with an average annual growth rate of 0.4% and an average annual growth area of 3929 ha.

### 3.2. Validation and Comparison of Forest Disturbance Detection Results

This study compared the forest disturbance results with the existing GFC products. Since GFC products only record the latest forest-loss time, this study extracted Nanning forest-loss products from GFC products for 2016 to 2020 (Figure 7). Combined with the forest-loss map created in this study (Figure 8), a total of 61 polygons were collected for change detection accuracy validation. These 61 polygons were forest-loss samples with a total of 20,875 pixels, including 18,031 loss pixels detected in this study and 12,566 loss pixels detected in GFC products. Overall, the detection accuracy of the breakpoint algorithm based on Landsat segments was 86.4%, while the accuracy of GFC forest change products was about 60%. Apparently, the disturbance result from the CCDC-STBR approach has higher accuracy than the GFC product. In the forest-loss maps, the spots detected by this study were more concentrated than those of GFC products. Still, on the whole, the regional loss distributions of the two are consistent are mainly concentrated in the middle and northwestern areas of Nanning.

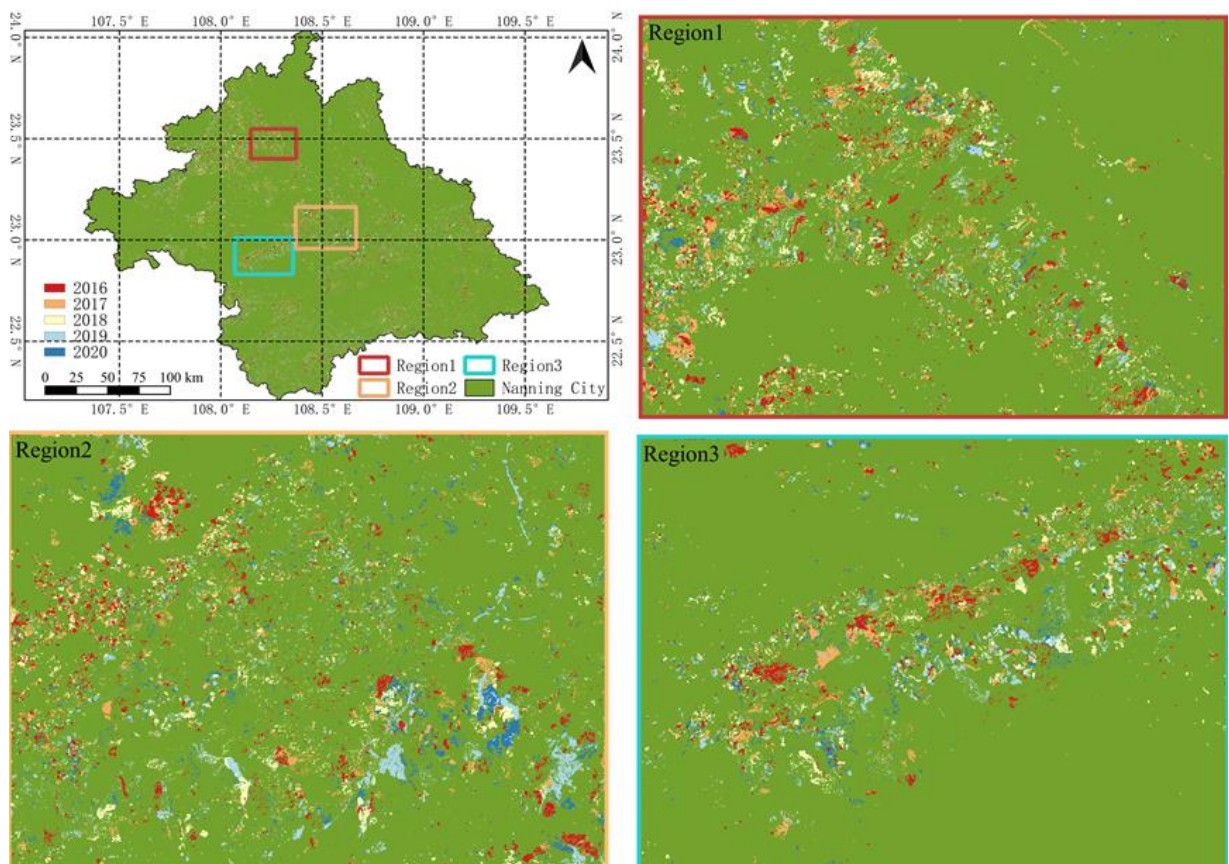

**Figure 7.** Global forest-loss maps of Nanning, 2016–2020.

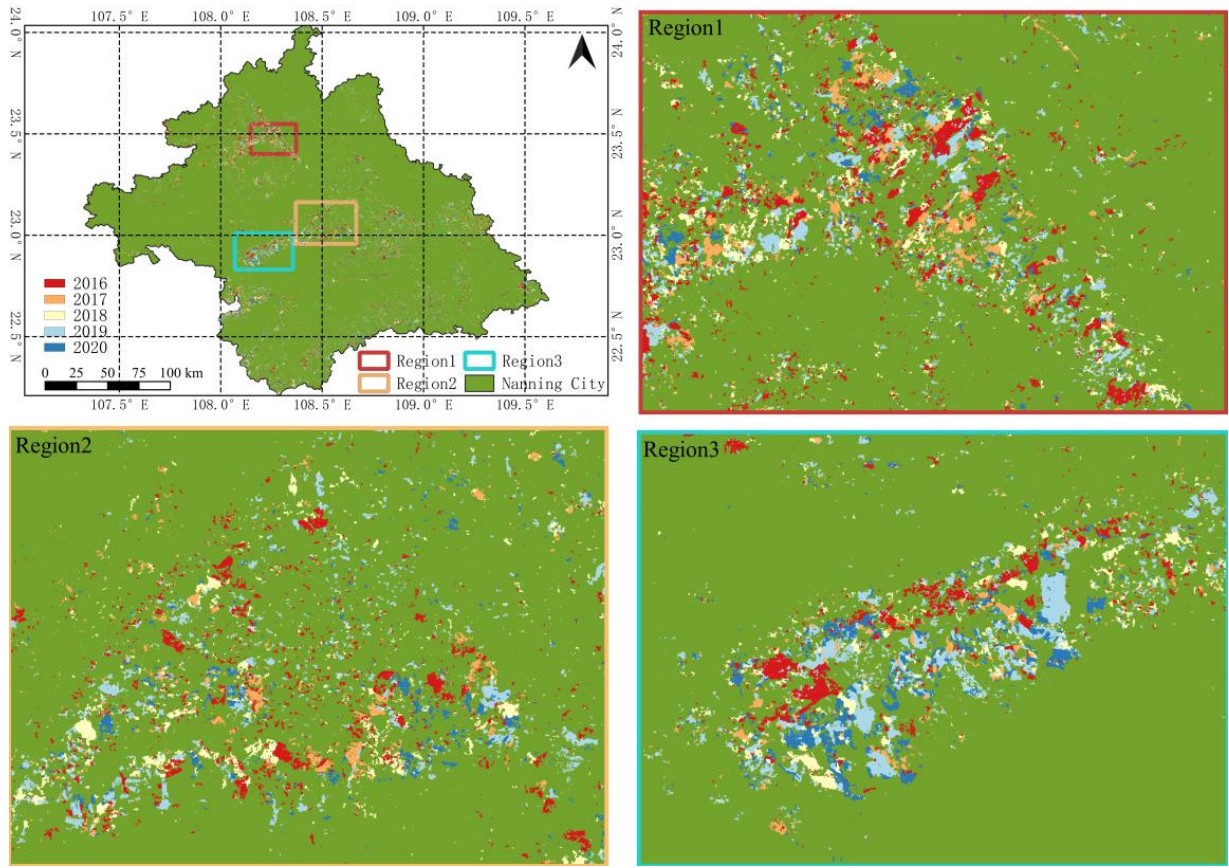

**Figure 8.** Detected forest-loss maps of Nanning, 2016–2020.

### 3.3. Forest Disturbance Results

The breakpoint change factor can measure the intensity of forest disturbance. In this paper, the sum of the *NDFI*, NBR, and NDVI confidences is defined as sum_confidence. The sum_confidence from 1987 to 2020 was calculated, and all the confidence values in Nanning are shown in the histogram (Figure 9). The histogram shows that most breakpoint confidence values are >1 and are mainly 0.8–2.8. A value of 1 means that two of the three index factors (*NDFI*, NBR, NDVI) are very likely to indicate that the point is a breakpoint. When the magnitude of a single index is three times that of the *RMSE*, its corresponding confidence value is exactly 0.5.

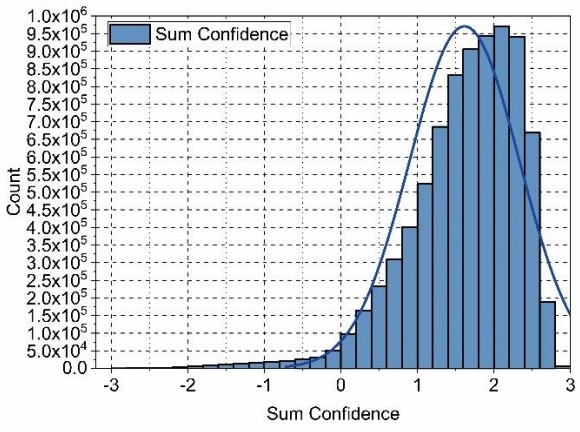

**Figure 9.** Histogram of sum_confidence in Nanning.

The center of the probability density curve fitted to the histogram of confidence values is at 1.5; that is, almost three index factors determine that the landcover type changed abruptly. The distribution of the breakpoint degree factor shows that there was relatively severe forest disturbance in Nanning from 1987 to 2020. This is directly related to the repeated rotations of plantation forest harvesting in Nanning, which is an intense type of forest disturbance.

First, breakpoints with the greatest sum confidence value of NBR, NDVI, and *NDFI* from Landsat segments were selected year by year; however, there was still salt and pepper noise in the breakpoint. Therefore, the existing morphological closed operation and post-processing algorithms, such as object-oriented connectivity detection and unsupervised clustering, were combined to filter out the noise of the breakpoints and make annual forest disturbance maps. This study also statistics the forest disturbance area from 1987 to 2020 (Figure 10). From the statistical results, intermittent deforestation is common in Nanning. Analysis of the annual forest-loss area shows that the forest disturbance area has increased and decreased intermittently, with periodic logging occurring every 2–8 years. In the monitored disturbance area, there were strong forest disturbances in 2002, 2005, and 2011, and each forest disturbance area was >200 ha. This intermittent large-scale forest disturbance is closely related to Nanning's main industry, timber plantations. In Nanning, fast-growing commercial forests have always been regarded as an important industry. Therefore, intermittent large-scale forest disturbance due to harvesting is normal. To further determine the spatial distribution of forest disturbance in Nanning, this study made annual forest disturbance maps (Figure 11).

Region1, Region2, and Region3 were selected, which had relatively concentrated forest-loss events. According to the map of forest disturbance from 1987 to 2020 (Figure 11), disturbed areas were mainly distributed in the central, eastern, and northwestern regions of Nanning. Disturbed areas were relatively concentrated and had obvious patchiness. The distributions of forest disturbance are also very regular in time and space. That is, a large area within the scope of every five years, different regions have different disturbance years, which is related to the planting and periodic harvesting of timber plantations.

To further detect the frequency of forest loss in Nanning City, the disturbance frequency was calculated from 1987 to 2020 (Figure 12). Frequency forest-loss events mainly occurred in the central, northern, and southwestern parts of Nanning. The frequency of forest disturbance in the forest hinterland of southwest China was the highest. In this region, the forest-loss event times had a dense spatial distribution. In addition, from 1987 to 2020, there were generally fewer than five forest-loss events in Nanning. That is, the average rotation period of plantation harvesting in Nanning was generally more than seven years.

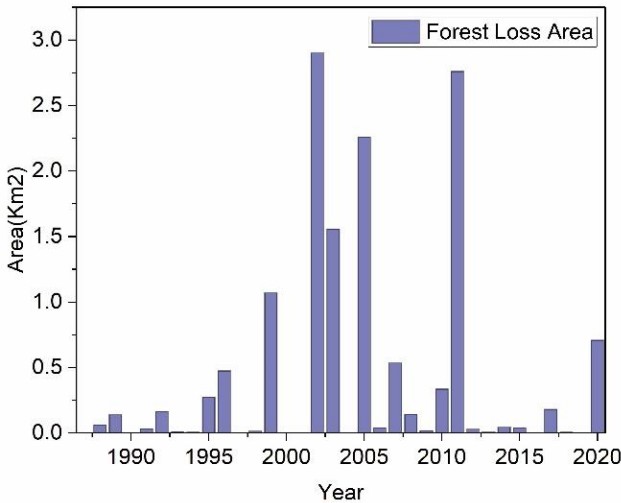

**Figure 10.** Forest-loss area in Nanning from 1987 to 2020.

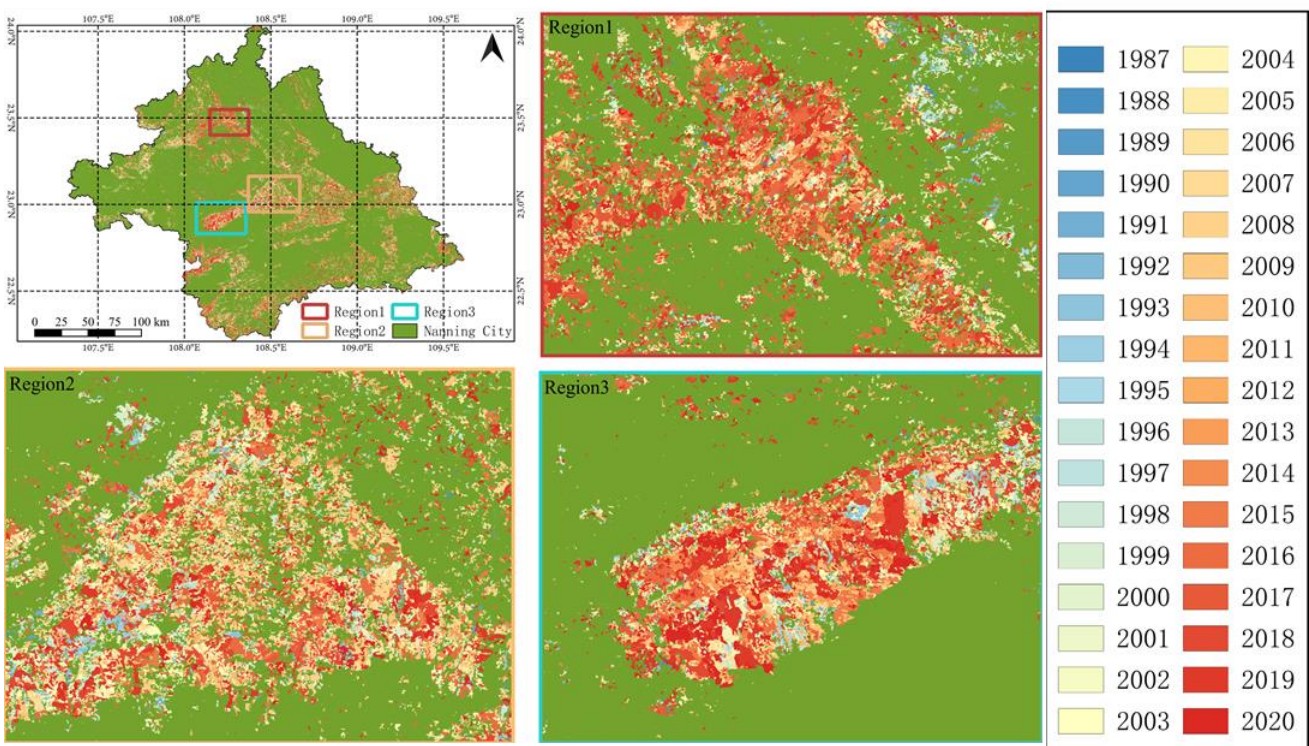

**Figure 11.** Forest loss in Nanning from 1987 to 2020.

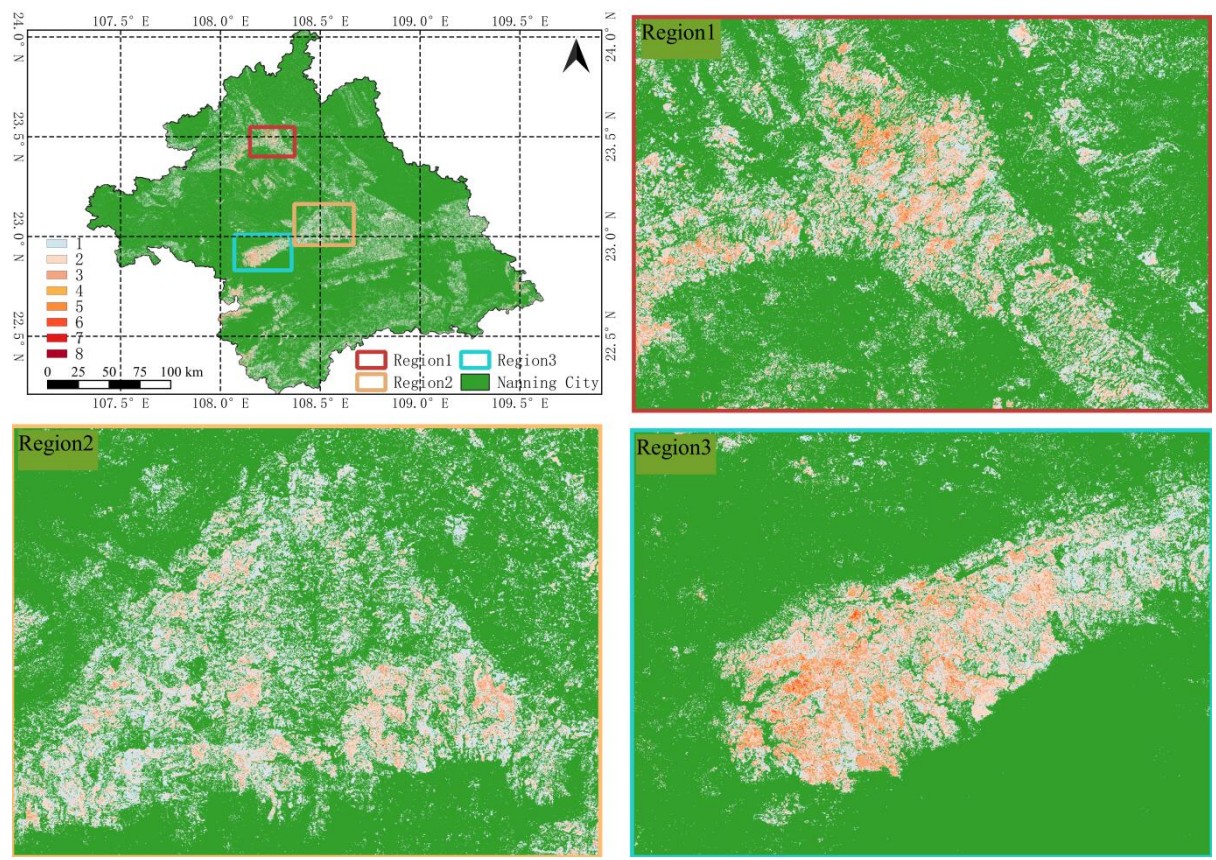

**Figure 12.** Spatial distribution of forest disturbance times from 1987 to 2020 in Nanning City.

One of the advantages of the CCDC-STBR algorithm used in this study is that this algorithm can acquire accurate annual durations of forest disturbance. The annual forest disturbance time from 1987 to 2020 was analyzed (Figure 13). The forest disturbance months in Nanning were mainly concentrated from July to December. September to December is the peak period of forest disturbance in Nanning, while the fewest disturbance events occur in January. From July, the frequency of forest disturbance events increases and peaks in November, which corresponds to plantation harvesting. Guangxi Province has abundant water, heat, and energy, and November is the peak period of plantation growth.

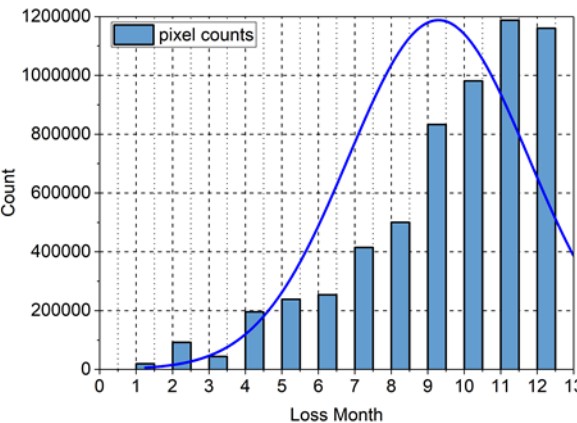

**Figure 13.** Forest-loss months from 1987 to 2020.

## 4. Discussion

The CCDC-STBR algorithm also has the advantages of the CCDC algorithm:

1. A harmonic model is used to fit the time-series change characteristics of each band, which can reduce the time-phased noise caused by periodic changes [35,36];
2. The algorithm is fully automated, and no empirical or global thresholds need to be specified in the detection process;
3. The algorithm can diagnose both interannual and intra-annual trends.

However, this algorithm also has some deficiencies. Its advantages and defects are discussed from the two aspects of land cover classification and disturbance detection.

### 4.1. Forest Distribution Classification

The feature subset optimization method used in this study can maximize land cover classification accuracy. Using seven bands of Landsat imagery (NIR, red, green, blue, Swir1, Swir2, temp), three indices (NBR, NDVI, *NDFI*), and auxiliary data (such as elevation, aspect, DEM, rainfall, tree cover), the classification algorithm can achieve a classification accuracy of up to 95.16%, which is higher than using other combinations of classification features. The classification accuracy of forest landcover is very high, which proves that multi-dimensional spectral-temporal change information is useful for distinguishing subtle differences between landcover types [37]. It also showed that the CCDC algorithm has the potential to overcome the low frequency of Landsat observations by using every observation on a per-pixel basis to build stable season-trend models; thus, it is suitable for forest monitoring in regions such as Nanning, which has cloudy and rainy climates.

Although the accuracy of forest cover classification based on the CCDC algorithm is high, there are still some deficiencies. The CCDC Landsat segment classification uses stable landcover as training samples, which increases the sampling selection complexity. In addition, the CCDC algorithm needs sufficient cloud-free observations to initialize the harmonic model, and some pixels will be missed if the number of observations in a year is less than the minimum observations set for the harmonic model [26]. The classification result for Nanning in 1987 had an unusually low forest coverage area because of insufficient observations.

*4.2. Forest Disturbance Detection Based on CCDC-STBR Algorithm*

This study uses the spectral trajectory breakpoint recognition algorithm to monitor the forest disturbance in Nanning year by year based on a harmonic model. Compared with LandTrendr, VCT, Verdet, and other change detection algorithms, this method can monitor the real-time forest disturbance to forest within a year on a monthly scale.

Omission errors exist in forest disturbance data; however, it is much better than that of GFC products, and there are several causes that resulted in the omission errors. Firstly, although the harmonic model greatly improves the continuity of time-series data, the cloudy and rainy climate of Nanning reduces the number of Landsat time-series observations. The low annual data acquisition rate and continuous pixel losses in the time series are both great challenges for the harmonic model to generate spectral-temporal coefficients. Secondly, the breakpoint detection method was adopted for change detection in this paper, which is not affected by classification accuracy. However, it is unreasonable to use the same statistical method to identify outliers of different indices because each index has a different sensitivity to forest disturbance events such as fire [10,38]. Third, at least six cloud-free observations are required to initialize the CCDC model in this paper. If the observation situation is not ideal and there are fewer than six cloud-free observations in a year, the omission error will increase. Apparently, there are quite a few areas in Nanning where the annual average number of Landsat observations is less than 6 from 1987 to 1995, but for other years, the annual average number of Landsat observations is sufficient (Figure 14). Fourth, some partial changes or observation noise can affect the initialization of the harmonic model [39]. The last, although the super-pixel clustering algorithm based on simple non-iterative clustering can eliminate some small pixel patches, partial changes are easily ignored [40].

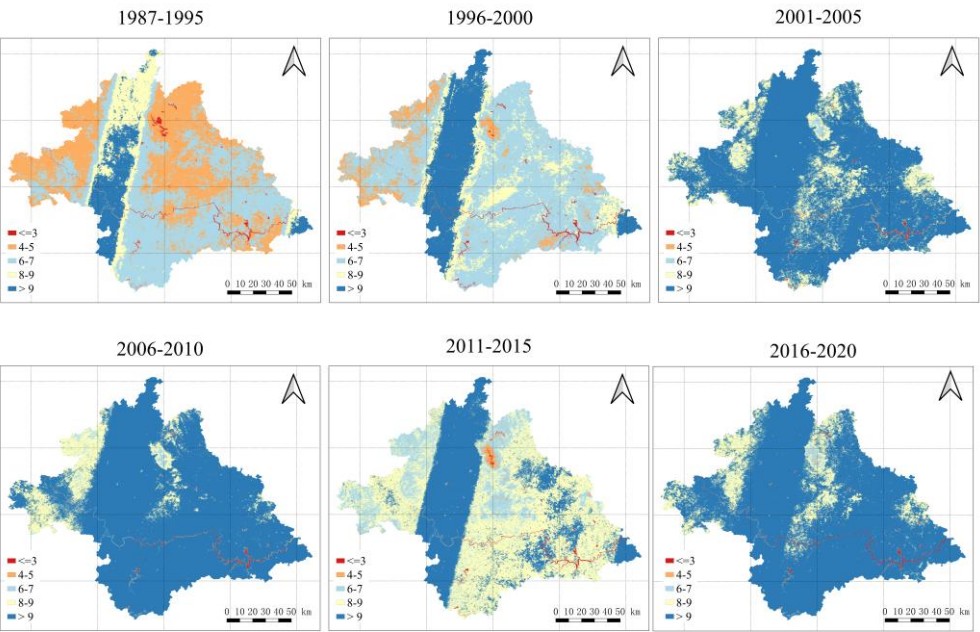

**Figure 14.** Annual average number of Landsat observations in the certain year range.

To extend the CCDC-STBR algorithm to regional or global scales, improve the accuracy and scope of its application to landcover classification, and to further improve the detection accuracy of forest-loss events, the following improvements are planned: (1) To evaluate the sensitivity of different combinations of forest change detection indices (such as different combinations of the normalized combustion index, normalized vegetation index, *NDFI*, and other indexes), statistical methods were used to detect and evaluate the breakpoint sensitivity. The most sensitive index combination was selected as the detection index of the timing breakpoint. (2) When using the post-processing algorithm, the breakpoint detection

results can be further combined with the confidence threshold and superpixel clustering algorithm to reduce the influence of salt and pepper noise and improve the accuracy of change detection. (3) The existing CCDC-STBR algorithm still has room for improvement in the statistical judgment of breakpoints, and different threshold standards can be set for different indices.

The CCDC-STBR algorithm can also be applied to detect the causes of forest disturbance for its abundant spectral-temporal change information from the harmonic model, such as fire, insects, and logging [41]. It also has the potential to be applied to detect vegetation phenology and forest degradation [41,42].

## 5. Conclusions

The forest monitoring method in this paper is an integration of existing classification comparison and trajectory tracking methods. Improvements for classification and spectral trajectory tracking are proved to be effective. For the classification results, the high classification accuracy showed that the optima subset feature selection method is effective in improving the classification accuracy. The result of optima selection shows that not the more classification features, the higher classification accuracy.

The spectral trajectory breakpoint recognition approach has the following advantages compared to the existing forest disturbance detection methods and the GFC product: (1) the CCDC-STBR algorithm can accurately capture the specific disturbance time, and we draw the conclusion that the forest disturbance time mainly concentrated in July to December; (2) the CCDC-STBR algorithm is fully automated, with no empirical threshold requirement and easy to be used on GEE; (3) the continuous trajectory breakpoint method has apparently higher disturbance detection accuracy than the GFC product, with *overall accuracy* up to 86.4%. It is also suitable for high-resolution time-series data such as Sentinel. Although the CCDC-STBR has a high requirement for computing resources, its integration with GEE is beneficial for its use at a larger spatial scale.

The forest monitoring in Nanning represents that: (1) from 1987 to 2020, Nanning continues to turn green [43], and the rate of this greening has increased rapidly since 2005, which is related to a series of new policies issued by the Nanning municipal government, such as closing mountains for afforestation, reasonable allocation of forest species, and rocky desertification integrated rehabilitation [34]; (2) the intermittent harvesting strategy affects the spatial and temporal distribution of forest disturbance, most forest areas have a disturbance frequency of about 7 years, and disturbance years in Nanning are spatially complementary. Nanning has abundant water and warm temperatures, and a series of forest management and ecological governance policies by the Nanning municipal government have contributed to the continuous development of local forestry and the ecological environment.

Comprehensively, the CCDC-STBR forest monitoring algorithm has the potential to be applied to global forest disturbance monitoring.

**Author Contributions:** Methodology, Y.Z., L.W. and Q.Z.; software, Y.Z. and F.T.; validation, Y.Z.; formal analysis, Y.Z., F.T. and B.Z.; investigation, Y.Z.; resources, Y.Z.; data curation, Y.Z.; writing—original draft preparation, Y.Z.; writing—review and editing, Y.Z., L.W., F.T., Q.Z., B.Z., N.H. and B.N.; visualization, Y.Z., F.T., Q.Z. and B.Z.; supervision, L.W.; project administration, L.W.; funding acquisition, L.W. All authors have read and agreed to the published version of the manuscript.

**Funding:** This research was supported by the National Key Research and Development Project of China (grant no. 2021YFE0117900) and the National Natural Science Foundation of China (grant no. 41871347).

**Institutional Review Board Statement:** Not applicable.

**Informed Consent Statement:** Not applicable.

**Acknowledgments:** We thank the USGS for distributing the Landsat time-series images. Furthermore, we thank the anonymous reviewers and members of the editorial team for their constructive comments. Thanks for their encouragement of all the authors and the master Xumiao Gao.

**Conflicts of Interest:** The authors declare no conflict of interest.

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
