# Peer review of "Continuous Change Detection and Classification—Spectral Trajectory Breakpoint Recognition for Forest Monitoring"

_land, doi:10.3390/land11040504_

Round 1

Reviewer 1 Report

The manuscript is devoted to application of Continuous Change Detection and Classification-Spectral Trajectory Breakpoint Recognition approach to forest disturbance monitoring including long-term trends. The approach is tested for forest cover in Nanjing City region from 1987 to 2020. The possibility of high-accuracy classification maps creation and forest disturbance detection proves the usefulness of approach application for its wider and further utilization as instrument in monitoring activity.

The research is well-documented and looks quite correct from the point of view of statistics.

The only remark for the manuscript – the use of abbreviations in abstract and key words for the paper. It is better to avoid as abstract must be readable for persons, who are not specialists in this subdivision of scientific branch.

The paper presents good income to the development of monitoring.

Reviewer 2 Report

The paper is very interesting. The authors present an approach Continuous Change Detection and Classification-Spectral Trajectory Breakpoint Recognition (CCDC-STBR) running on Google Earth Engine (GEE) for monitoring forest disturbance and forest long-term trends. They used this approach to monitor forest disturbance and the change of forest cover rate from 1987 to 2020 in Nanning City, China.

The proposed approach helps to generate high-accuracy classification maps, detect the forest disturbance time at a month scale, capture the thinning cycle of plantations.

The paper is well structured. The authors pursue the goal successfully.
I suggest you:

  • integrate possible developments of this research;
  • explain all the acronyms in the text.

Reviewer 3 Report

The research aims to present a continuous change detection and classification-spectral trajectory breakpoint recognition (CCDC-STBR) approach, running on Google Earth Engine (GEE), for monitoring forest disturbance and forest long-term trends. 
Overall, the research lack in novelty, and the paper was structured in an ambiguous and rather not-orthodoxal way.
In particular:

1. The aims are not clear and well-focused. Indeed, stating "monitor forest disturbance" means an incredible number of things covering different aspects of forest disturbance. As a matter of fact, this point remains unclear and obscure even throughout the entire paper, with particular reference for the "Results" and "Discussion" sections;

2. Several approaches have been proposed in the literature for monitoring forest disturbance. However, throughout the entire text, the authors missed explaining why and in which way their method differs;

3. Some statements are simply false and/or inaccurate. Here is an example "Of the terrestrial ecosystems, forest is predominant as it has the most extensive coverage area, broadest distribution, most complex compositional structure, richest biodiversity, and highest primary productivity." Please note that soil is the predominant one of all terrestrial ecosystems;

4. Introduction is too long with a lot of ancillary not necessary information. The authors fail to stay focused on the primary papers' aspects;

5. Results and Discussion are characterized for overuse of speculative comments based on the author's suggestions rather than on fact;

6. Conclusions are almost two pages of repetitive comments already reported in the previous sections, thus being inconclusive.

For all of the previous, rejection is suggested.

Round 2

Reviewer 3 Report

Authors did several changes. However, this paper nothing add in terms of novelty in this field of research.